# Current Knowledge in Skin Metabolomics: Updates from Literature Review

**DOI:** 10.3390/ijms23158776

**Published:** 2022-08-07

**Authors:** Alessia Paganelli, Valeria Righi, Elisabetta Tarentini, Cristina Magnoni

**Affiliations:** 1Clinical and Experimental Medicine Ph.D. Program, University of Modena and Reggio Emilia, 41124 Modena, Italy; 2Regenerative and Oncological Dermatological Surgery Unit, Modena University Hospital, 41124 Modena, Italy; 3Department for Life Quality Studies, University of Bologna, 47921 Rimini, Italy; 4Servizio Formazione, Ricerca e Innovazione, Modena University Hospital, 41124 Modena, Italy

**Keywords:** skin metabolomics, dermatology, biomarkers

## Abstract

Metabolomic profiling is an emerging field consisting of the measurement of metabolites in a biological system. Since metabolites can vary in relation to different stimuli, specific metabolic patterns can be closely related to a pathological process. In the dermatological setting, skin metabolomics can provide useful biomarkers for the diagnosis, prognosis, and therapy of cutaneous disorders. The main goal of the present review is to present a comprehensive overview of the published studies in skin metabolomics. A search for journal articles focused on skin metabolomics was conducted on the MEDLINE, EMBASE, Cochrane, and Scopus electronic databases. Only research articles with electronically available English full text were taken into consideration. Studies specifically focused on cutaneous microbiomes were also excluded from the present search. A total of 97 papers matched all the research criteria and were therefore considered for the present work. Most of the publications were focused on inflammatory dermatoses and immune-mediated cutaneous disorders. Skin oncology also turned out to be a relevant field in metabolomic research. Only a few papers were focused on infectious diseases and rarer genetic disorders. All the major metabolomic alterations published so far in the dermatological setting are described extensively in this review.

## 1. Introduction

Metabolomics is an emerging science focused on the study of the global metabolic profile of cells, tissues, organs, or even entire organisms by systematic identification and quantitation of all metabolites in such biological systems [1]. Metabolites are low-molecular-weight (<1500 Da) biomolecules that participate in metabolic reactions in a cell or in the organism where their total collection constitutes the metabolome [2]. Metabolites are the downstream products of gene transcription and vary in relation to any possible internal or external stimulus (e.g., genetic mutations), therefore the presence of a specific metabolic pattern can be closely related to a physiological or pathological phenotype [3]. 

Targeted and untargeted approaches are possible in metabolomics, the former focusing on the analysis of a subset of known compounds, and the latter focusing on the whole array of metabolites detected. Using both approaches, hundreds to thousands of metabolites can be detected (but not necessarily identified or quantified). Obtained data are usually analyzed following standardized metabolomics pipelines [4] and information is extracted using state-of-the-art statistical tools [5]. 

Various innovative and advanced analytical techniques are currently employed for the detection and quantification of metabolites and metabolite-related metabolic pathways, including mass spectrometry (MS), nuclear magnetic resonance (NMR), Fourier transform infrared (FT-IR) spectroscopy and Raman spectroscopy, and separation techniques like ion mobility systems (IMS), capillary electrophoresis (CE) systems, gas chromatography (GC), and liquid chromatography (LC) systems, which are frequently combined with each other [6,7]. Whatever the chosen method, the typical workflow for metabolomics starts from sample collection and leads to a large volume of data composed of many metabolites/compounds identified with absolute or relative concentrations (see Figure 1). Finally, ANOVA, multivariate statistical analysis, and clustering/classification analysis (such as Principal Component Analysis (PCA) and Partial Least Squares Discriminant Analysis (PLS-DA)) allow the selection of the principal metabolites and metabolic pathways involved in a specific setting [8]. Furthermore, the development of metabolic database analytical measurement systems and data analysis techniques have increased rapidly, improving the identification, evaluation, and accurate assignment of measured metabolites to the corresponding metabolic pathways. The combination of approaches with measurement methods based on GC-MS, HLPC-MS, and NMR have enabled a significant expansion of metabolic databases on Internet platforms, such as the Kyoto Encyclopedia of Genes and Genomes (KEGG) and the Human Metabolome Database (HMDB), in recent years. Metabolomics has various applications in biomedical research, considering the important role of metabolites in many therapeutic and pathological processes. More than 95% of the assays currently available in the clinical practice use metabolites for diagnostic purposes (e.g., glucose for diabetes). In fact, metabolite alterations characterize many diseases, either as a cause or as a result of the disease itself [6]. 

As for other medical settings, dermatological metabolomics can help to better understand and explore the intimate link between external stimuli and the internal responses of the organism, therefore potentially revealing new diagnostic and/or therapeutic scenarios. 

Skin metabolites can originate both from skin cellular components (e.g., keratinocytes, melanocytes, fibroblasts) or from cutaneous fluids, such as sweat, sebum, and interstitial fluid, and can be produced in response to environmental stressors, drugs, or genetic mutations. Skin metabolomic studies can therefore provide useful metabolic biomarkers for the diagnosis, prognosis, and therapy of both localized and systemic dermatological disorders [9]. For example, amino acids are considered key metabolites for wound healing, acid–base balance and water retention, protection against sunlight damage, and maintenance of the skin microbiome [10]. 

Skin metabolites can also provide suitable information about a broad spectrum of non-dermatological diseases. With regards to this, Moraes et al. identified the 4-Hydroxy-4-methylpentan-2-one, nonanal, and toluene from skin volatiles samples as biomarkers of malaria [11]. Calderon-Santiago et al. recognized non-anedioic or azelaic acid, monoglyceride MG (22:2), suberic acid, a trihexose, and a tetrahexose from human sweat as potential biomarkers for lung cancer screening [12]. 

On the other hand, dermatological systemic disorders can sometimes be characterized by alterations in the metabolomics of biofluids (blood, urine). Armstrong and co-authors, for example, analyzed serum metabolites of patients affected by psoriasis and/or psoriatic arthritis, and identified various biomarkers that could help discriminate these two categories, such as alpha ketoglutaric acid and lignoceric acid variations. Moreover, a metabolomics analysis of skin tissues and biofluids has also been employed for the assessment of toxic and non-toxic external substance exposure. For example, Lee et al. analyzed the association of propyl paraben exposure with aeroallergen sensitization using urine metabolomics [13], while our group evaluated metabolomic pattern variations in patients with actinic keratoses before and after field-cancerization therapy with ingenol mebutate [14]. 

The main goal of the present review is to give a comprehensive overview of the published studies involving metabolomics in the dermatological setting. 

## 2. Results and Discussion

The bibliographic research identified 441 publications, with 279 papers being considered after duplicate removal. Five papers were excluded because no full text in English was available. 

No complete fulfillment of the other inclusion criteria was found in 177 papers. A total of 97 papers matched all the research criteria and were therefore considered for the present work (for PRISMA Flowchart see Figure 2; the complete list of the papers can be found in Table 1). 

Most of the publications were focused on dermatological disorders rather than on skin function assays (70 vs. 23, *p* < 0.05). 

The vast majority of the publications were focused on inflammatory dermatoses and immune-mediated disorders. Another significant part of dermatological literature on metabolomics regarded skin oncology, while only a few papers were published on infectious diseases and rarer genetic disorders. 

While the shared aim of the 93 collected papers was to identify potential alterations of skin metabolites, this objective was achieved through a wide range of possibilities. 

As for the analytical techniques employed, in fact, LC-MS based publications accounted for more than 50% (*n* = 49) of the total, with LC-MS certainly being the most frequently used technique for cutaneous metabolome assessment [15,16,17,18]. LC-MS-based methods were followed by NMR and GC-MS. Only a few research articles used other methods, such as CE-TOFMS or FI-TOFMS. In 17 cases (out of 93), combinations of different metabolomic methods were employed, with LC-MS and GC-MS being coupled in 5 works.

The main dermatoses are now discussed in more detail for their principal metabolomics alterations.

**Table 1 ijms-23-08776-t001:** List of the 97 papers retrieved with our search. Authors have been listed in alphabetical order. Journal names are abbreviated according to MedLine nomenclature.

Author	Year	Journal	Article Title	Ref.
Abaffy et al.	2013	*Metabolomics*	Comparative analysis of volatile metabolomics signals from melanoma and benign skin: a pilot study	[19]
Abaffy et al.	2011	*J Cancer Sci Ther.*	A case report—Volatile metabolomic signature of malignant melanoma using matching skin as a control	[20]
Abaffy et al.	2010	*PLoS ONE*	Differential Volatile Signatures from Skin, Naevi and Melanoma: A Novel Approach to Detect a Pathological Process	[21]
Acharjee et al.	2021	*Am J Transl Res.*	Multi-omics-based identification of atopic dermatitis target genes and their potential associations with metabolites and miRNAs.	[22]
Afghani et al.	2021	*Metabolites*	Enhanced Access to the Health-Related Skin Metabolome by Fast, Reproducible and Non-Invasive WET PREP Sampling	[18]
Al-Mubarak et al.	2011	*PLoS Negl Trop Dis*	Serum Metabolomics Reveals Higher Levels of Polyunsaturated Fatty Acids in Lepromatous Leprosy: Potential Markers for Susceptibility and Pathogenesis	[23]
Alkhalil et al.	2020	*J Burn Care Res*	Cutaneous Thermal Injury Modulates Blood and Skin Metabolomes Differently in a Murine Model	[24]
Armstrong et al.	2014	*F1000Res*	Metabolomics in psoriatic disease: pilot study reveals metabolite differences in psoriasis and psoriatic arthritis	[25]
Ashrafi et al.	2020	*PLoS ONE*	A microbiome and metabolomic signature of phases of cutaneous healing identified by profiling sequential acute wounds of human skin: An exploratory study.	[10]
Bai et al.	2019	*EBioMedicine*	Identification of a natural inhibitor of methionine adenosyltransferase 2A regulating one-carbon metabolism in keratinocytes.	[26]
Bengtsson et al.	2016	*PLoS ONE*	Metabolic Profiling of Systemic Lupus Erythematosus and Comparison with Primary Sjögren’s Syndrome and Systemic Sclerosis	[27]
Cappellozza et al.	2021	*Microsc Microanal*	Integrated Microscopy and Metabolomics to Test an Innovative Fluid Dynamic System for Skin Explants In Vitro	[28]
Carrola et al.	2016	*Nanotoxicology*	Metabolomics of silver nanoparticles toxicity in HaCaT cells: Structure-activity relationships and role of ionic silver and oxidative stress	[29]
Chao et al.	2017	*Phytomedicine*	Melaleuca quinquenervia essential oil inhibits α-melanocyte-stimulating hormone-induced melanin production and oxidative stress in B16 melanoma cells.	[30]
Chen et al.	2021	*Theranostics*	Metabolomic profiling reveals amino acid and carnitine alterations as metabolic signatures in psoriasis	[31]
Chen et al.	2021	*J Invest Dermatol*	Measurement of Melanin Metabolism in Live Cells by [U-13 C]-L-Tyrosine Fate Tracing Using Liquid Chromatography-Mass Spectrometry.	[32]
Cheng et al.	2020	*Biomed Chromatogrs*	Spleen and thymus metabolomics strategy to explore the immunoregulatory mechanism of total withanolides from the leaves of Datura metel L. on imiquimod induced psoriatic skin dermatitis in mice	[33]
Cheng et al.	2020	*J Pharm Biomed Anal*	Integrated serum metabolomics and network pharmacology approach to reveal the potential mechanisms of withanolides from the leaves of Datura metel L. on psoriasis.	[34]
Dutkiewicz et al.	2016	*Clin Chem*	Hydrogel Micropatch and Mass Spectrometry–Assisted Screening for Psoriasis-Related Skin Metabolites	[35]
Elbayed et al.	2013	*Chem Res Toxicol*	HR-MAS NMR Spectroscopy of Reconstructed Human Epidermis: Potential for the in Situ Investigation of the Chemical Interactions between Skin Allergens and Nucleophilic Amino Acids.	[36]
Emmert et al.	2020	*Exp Dermatol*	Stratum corneum lipidomics analysis reveals altered ceramide profile in atopic dermatitis patients across body sites with correlated changes in skin microbiome.	[37]
Fedele et al.	2013	*Biomed Pharmacother*	Prognostic relationship of metabolic profile obtained of melanoma B16F10.	[38]
Fitzgerald et al.	2020	*J Proteome Res*	Host Metabolic Response in Early Lyme Disease	[39]
Frontiñán-Rubio et al.	2018	*Nanoscale*	Differential effects of graphene materials on the metabolism and function of human skin cells.	[40]
Fukumoto et al.	2017	*J Dermatol*	Novel serum metabolomics-based approach by gas chromatography/triple quadrupole mass spectrometry for detection of human skin cancers: candidate biomarkers.	[41]
Gao et al.	2012	*Anal Bioanal Chem*	A reversed-phase capillary ultra-performance liquid chromatography-mass spectrometry (UPLC-MS) method for comprehensive top-down/bottom-up lipid profiling	[17]
Harker et al.	2014	*J Dermatol Sci*	Functional characterisation of a SNP in the ABCC11 allele—Effects on axillary skin metabolism, odour generation and associated behaviours	[42]
Hashimoto et al.	2019	*Pharm Res*	Metabolome Analysis Reveals Dermal Histamine Accumulation in Murine Dermatitis Provoked by Genetic Deletion of P-Glycoprotein and Breast Cancer Resistance Protein	[43]
Hellmann et al.	2018	*J Invest Dermatol*	Biosynthesis of D-series resolvins in skin provides insights into their role in tissue repair	[44]
Hollywood et al.	2015	*Mol Biosyst*	Exploring the mode of action of dithranol therapy for psoriasis: a metabolomic analysis using HaCaT cells.	[45]
Hooton, Li.	2017	*Anal Chem*	Nonocclusive Sweat Collection Combined with Chemical Isotope Labeling LC-MS for Human Sweat Metabolomics and Mapping the Sweat Metabolomes at Different Skin Locations.	[46]
Hosseini et al.	2018	*Cell Rep*	Energy Metabolism Rewiring Precedes UVB-Induced Primary Skin Tumor Formation	[47]
Huang et al.	2014	*J Proteome Res*	Serum Metabolomics Study and Eicosanoid Analysis of Childhood Atopic Dermatitis Based on Liquid Chromatography-Mass Spectrometry	[48]
Ilves et al.	2021	*Acta Derm Venereol*	Metabolomic Analysis of Skin Biopsies from Patients with Atopic Dermatitis Reveals Hallmarks of Inflammation, Disrupted Barrier Function and Oxidative Stress.	[49]
Jacob et al.	2019	*Metabolites*	Metabolomics Distinguishes DOCK8 Deficiency from Atopic Dermatitis: Towards a Biomarker Discovery	[50]
Jacques et al.	2021	*Arch Toxic*	Safety assessment of cosmetics by read across applied to metabolomics data of in vitro skin and liver models	[51]
Jansen et al.	2013	*PNAS*	ABCC6 prevents ectopic mineralization seen in pseudoxanthoma elasticum by inducing cellular nucleotide release	[52]
Jiang, Kang, Yu.	2017	*J Chromatogr B Analyt Technol Biomed*	Cross-platform metabolomics investigating the intracellular metabolic alterations of HaCaT cells exposed to phenanthrene	[53]
Jung et al.	2019	*Sci Rep*	Seven-day Green tea Supplementation Revamps Gut Microbiome and caecum/Skin Metabolome in Mice from Stress	[54]
Kaiser et al.	2021	*JMIR Res Proto*	Multiscale Biology of Cardiovascular Risk in Psoriasis: Protocol for a Case-Control Study	[55]
Kamleh et al.	2015	*J. Proteome Res.*	LC-MS Metabolomics of Psoriasis Patients Reveals Disease Severity Dependent Increases in Circulating Amino Acids That Are Ameliorated by Anti-TNFα Treatment.	[56]
Kang et al.	2017	*Br J Dermatol.*	Exploration of candidate biomarkers for human psoriasis based on gas chromatography-mass spectrometry serum metabolomics.	[57]
Khandelwal et al.	2014	*J Lipid Res*	1H NMR-based lipidomics of rodent fur: species-specific lipid profiles and SCD1 inhibitor-related dermal toxicity.	[58]
Khosravi et al.	2019	*Mol Med.*	Active repurposing of drug candidates for melanoma based on GWAS, PheWAS and a wide range of omics data.	[59]
Kim et al.	1989	*J Invest Dermatol*	1H NMR Spectroscopy: An Approach to Evaluation of Diseased Skin In Vivo.	[60]
Kishikawaa et al.	2021	*Journal of Dermatological Science*	Large-scale plasma-metabolome analysis identifies potential biomarkers of psoriasis and its clinical subtypes	[61]
Kosmopoulou et al.	2020	*Int. J. Mol. Sci.*	Human Melanoma-Cell Metabolic Profiling: Identification of Novel Biomarkers Indicating Metastasis.	[62]
Kuehne et al.	2015	*Mol Cell*	Acute Activation of Oxidative Pentose Phosphate Pathway as First-Line Response to Oxidative Stress in Human Skin Cells	[63]
Kuehne et al.	2017	*BMC Genomics*	An integrative metabolomics and transcriptomics study to identify metabolic alterations in aged skin of humans in vivo.	[64]
Le et al.	2018	*J Mass Spectrom.*	Accelerated, untargeted metabolomics analysis of cutaneous T-cell lymphoma reveals metabolic shifts in plasma and tumor adjacent skins of xenograft mice.	[65]
Lee et al.	2021	*Scientific Reports*	The potential pathways underlying the association of propyl paraben exposure with aeroallergen sensitization and EASI score using metabolomics analysis.	[13]
Li, Wei, Kuang.	2021	*J Pharm Biomed Anal*	UPLC-orbitrap-MS-based metabolic profiling of HaCaT cells exposed to withanolides extracted from Datura metel.L: Insights from an untargeted metabolomics.	[66]
Li et al.	2021	*Front Pharmacol*	Pithecellobium clypearia: Amelioration Effect on Imiquimod-Induced Psoriasis in Mice Based on a Tissue Metabonomic Analysis	[67]
Liang, Zhang, Cai.	2021	*Sci Total Environ*	New insights into the cellular mechanism of triclosan-induced dermal toxicity from a combined metabolomic and lipidomic approach	[68]
Liu et al.	2020	*Cells*	(R)-Salbutamol Improves Imiquimod-Induced Psoriasis-Like Skin Dermatitis by Regulating the Th17/Tregs Balance and Glycerophospholipid Metabolism.	[69]
Lutz et al.	2007	*Wound Rep Reg*	Conditions of wound healing and cutaneous growth affect metabolic performance of skin following plastic surgery	[70]
Malvi et al.	2021	*Mol Metab*	N-acylsphingosine amidohydrolase 1 promotes melanoma growth and metastasis by suppressing peroxisome biogenesis-induced ROS production	[71]
Marathe et al.	2021	*J Invest Dermatol*	Multi-omics analysis and systems biology integration identifies the roles of IL-9 in keratinocyte metabolic reprogramming.	[72]
Mayboroda et al.	2016	*Int J Infect Dis*	Exploratory urinary metabolomics of type 1 leprosy reactions	[73]
Mendez et al.	2020	*In Vitro Cel Dev Biol Anim*	Delineating cell behavior and metabolism of non-melanoma skin cancer in vitro.	[74]
Misra et al.	2021	*Sci Rep*	Multi-omics analysis to decipher the molecular link between chronic exposure to pollution and human skin dysfunction	[75]
Molins et al.	2017	*Sci Transl Med.*	Metabolic Differentiation of Early Lyme Disease from Southern Tick-Associated Rash Illness (STARI).	[76]
Mora-Ortiz et al.	2019	*Metabolomics*	Thanatometabolomics: introducing NMR-based metabolomics to identify metabolic biomarkers of the time of death	[77]
Morvan, Cachin.	2022	*J Proteome Res*	Untargeted 2D NMR Metabolomics of [13C-methyl]Methionine-Labeled Tumor Models Reveals the Non-DNA Methylome and Provides Clues to Methyl Metabolism Shift during Tumor Progression	[78]
Mun et al.	2016	*PLoS ONE*	Discrimination of Basal Cell Carcinoma from Normal Skin Tissue Using High-Resolution Magic Angle Spinning 1H NMR Spectroscopy	[79]
Niang et al.	2015	*Sci Rep*	Metabolomic profiles delineate mycolactone signature in Buruli ulcer disease	[80]
Niedzwiecki et al.	2018	*Anal Chem*	Human Suction Blister Fluid Composition Determined Using HighResolution Metabolomics	[81]
Ottas et al.	2017	*Arch Dermatol Res*	The metabolic analysis of psoriasis identifies the associated metabolites while providing computational models for the monitoring of the disease.	[82]
Palacios-Ferrer et al.	2021	*Mol Oncol*	Metabolomic profile of cancer stem cell-derived exosomes from patients with malignant melanoma	[83]
Protsyuk et al.	2018	*Nat Protoc*	3D molecular cartography using LC–MS facilitated by Optimus and ‘ili software.	[16]
Rasmussen et al.	2016	*J Proteome Res*	Untargeted metabolomics analysis of ABCC6-deficient mice discloses an altered metabolic liver profile	[84]
Righi et al.	2021	*Cancers*	Metabolomic Analysis of Actinic Keratosis and SCC Suggests a Grade-Independent Model of Squamous Cancerization	[85]
Righi et al.	2019	*Sci Rep*	Field cancerization therapy with ingenol mebutate contributes to restoring skin-metabolism to normal-state in patients with actinic keratosis: a metabolomic analysis	[14]
Sahoo et al.	2017	*J Invest Dermatol*	MicroRNA-211 Regulates Oxidative Phosphorylation and Energy Metabolism in Human Vitiligo.	[86]
Santana-Filho et al.	2017	*Sci Rep*	NMR metabolic fingerprints of murine melanocyte and melanoma cell lines: application to biomarker discovery	[87]
Sarkar et al.	2017	*J Invest Dermatol*	Endogenous glucocorti-coid deficiency in psoriasis promotes inflammation and ab-normal differentiation	[88]
Schilf et al.	2021	*Int J. Mol Sci*	A Mitochondrial Polymorphism Alters Immune Cell Metabolism and Protects Mice from Skin Inflammation	[89]
Seo et al.	2020	*Molecules*	Metabolomics Reveals the Alteration of Metabolic Pathway by Alpha-Melanocyte-Stimulating Hormone in B16F10 Melanoma Cells.	[90]
Sitter et al.	2013	*BMC Dermatol*	Metabolic changes in psoriatic skin under topical corticosteroid treatment	[91]
Sood et al.	2017	*Wound Repair Regen*	Targeted metabolic profiling of wounds in diabetic and nondiabetic mice.	[92]
Sreedhar et al.	2019	*Proteomics*	UCP2 overexpression redirects glucose into anabolic metabolic pathways.	[93]
Tarentini et al.	2021	*Sci Rep*	Integrated metabolomic analysis and cytokine profiling define clusters of immuno-metabolic correlation in new-onset psoriasis	[94]
Taylor et al.	2020	*PLoS ONE*	Metabolomics of primary cutaneous melanoma and matched adjacent extratumoral microenvironment.	[95]
Tilton et al.	2015	*Toxicol Appl Pharmacol*	Data integration reveals key homeostatic mechanisms following low dose radiation exposure	[96]
Wang et al.	2018	*AMIA Annu Symp Proc*	Combining mechanism-based prediction with patient-based profiling for psoriasis metabolomics biomarker discovery.	[97]
Wei et al.	2019	*BMJ Open.*	The association of tryptophan and phenylalanine are associated with arsenic-induced skin lesions in a Chinese population chronically exposed to arsenic via drinking water: a case–control study.	[98]
Wild et al.	2021	*Acta Physiol (Oxf)*	Aestivation motifs explain hypertension and muscle mass loss in mice with psoriatic skin barrier defect.	[99]
Wilkins et al.	2021	*Metabolomics*	A comprehensive protocol for multiplatform metabolomics analysis in patient-derived skin fibroblasts.	[15]
Wooding et al.	2020	*Anal Bioanal Chem*	Chemical profiling of the human skin surface for malaria vector control via a non-invasive sorptive sampler with GC×GC-TOFMS	[100]
Yang et al.	2021	*Sci Rep*	Metabolomics study of fbroblasts damaged by UVB and BaP	[101]
Zeng et al.	2017	*Gigascience*	Lipidomics profiling reveals the role of glycerophospholipid metabolism in psoriasis.	[102]
Zhang et al.	2020	*J Ethnopharmacol*	NMR-based metabolomic analysis for the effects of Huiyang Shengji extract on rat diabetic skin ulcers.	[103]
Zhang et al.	2019	*Metabolomics*	Metabolomic profiling for identification of potential biomarker in patients with dermatomyositis	[104]
Zhou et al.	2017	*Oncotarget*	Integration of microRNAome, proteomics and metabolomics to analyze arsenic-induced malignant cell transformation	[105]
Zhu et al.	2021	*Front Pharmacol*	Integrated Proteomics and Metabolomics Link Acne to the Action Mechanisms of Cryptotanshinone Intervention	[106]
Zhu et al.	2020	*Molecules*	The Synthetic Flavonoid Derivative GL-V9 Induces Apoptosis and Autophagy in Cutaneous Squamous Cell Carcinoma via Suppressing AKT-Regulated HK2 and mTOR Signals	[107]
Zinkevičienė et al.	2016	*Int Arch Allergy Immunol*	Activation of Tryptophan and Phenylalanine Catabolism in the Remission Phase of Allergic Contact Dermatitis: A Pilot Study	[108]

### 2.1. Psoriasis

Most of the studies on psoriasis are based on the ex-vivo metabolomic analyses of human skin and/or in-vitro metabolomic analysis of blood samples. Aminoacidic metabolism seems to be impaired in the setting of psoriasis [31,60]. In particular, a reduction in terms of content in alanine, glutamine, and asparagine seems to be present in cutaneous and plasmatic samples from psoriatic patients, while taurine is reduced in psoriatic skin. On the other hand, an increase of GSH [94], methionine, and arginine were also observed in skin affected by psoriasis [35,88]. Lysophospahatidycholine (lysoPC) and inositol metabolism have also been found to be impaired in psoriasis; the hydrolysis products of LysoPC are involved in inflammatory processes [82]. Ottas and co-authors hypothesized that those alterations could be due to the higher demand for amino acids in the hyperproliferative epidermis where de novo synthesis of proteins is upregulated and the rate of mitosis in basal keratinocytes is increased compared to non-lesional skin, which is in line with previously published observations [56]. Aminoacidic variations present in psoriatic skin are not always found in serum samples from psoriatic patients. Our group, for example, previously demonstrated a significant increase in ascorbate and a decrease in scyllo-inositol in psoriatic skin, while dimethylglycine and isoleucine appeared to be increased in patient sera [94]. Not surprisingly, variations in metabolite levels are often related to cytokine imbalances. The integration of metabolic and immunological data was fundamental in this setting and led us to the identification of a psoriasis molecular signature composed of IL-6, IL1-ra, DMG, CCL4, Ile, Gly, and IL-8 [94].

Zeng et al. postulated a central role for glycerophospholipid metabolism in psoriasis based on lipidomics profiling [102]. A central role for glycerophospholipids and, more in general, lipidic metabolism in psoriasis pathogenesis was confirmed by subsequent works [33,34,66,69].

Psoriasis is notably not associated with DNA damage, and no changes in such metabolisms were also confirmed by several metabolomic studies [35,88,94].

Most of the available literature on psoriasis metabolomics aims at identifying potential diagnostic markers of the disease [57,61,97]. However, the effects of topical agents for the treatment of psoriasis have also been widely studied in the last decade [26,45,69,91]. Recently, growing interest has risen in steroid-sparing therapies for psoriasis, and the efficacy of several plant extracts (e.g., Pithecellobium clypearia, Datura metel) has been demonstrated by various authors though the use of metabolomic analyses [33,34,66,67].

Potentially, metabolomics can also be used as a tool for the detection of the presence of specific comorbidities; in fact, metabolomic changes typical of psoriatic arthritis and cardiovascular disorders have already been described in literature [25,55,99]. For example, relatively lower levels of alpha ketoglutaric acid and increased lignoceric acid characterize patients with arthritis compared to psoriatic subjects with cutaneous involvement only [25]. Metabolomic approaches also led to the discovery of circulatory adjustments in skin blood flow as a fundamental element for limiting the trans-epidermal water loss (TEWL) of psoriatic skin, but potentially resulting in arterial hypertension [55].

Interestingly, new non-invasive approaches have also been explored for metabolome assessment in the setting of psoriasis; Dutkiewicz and collaborators reported on the use of hydrogel micropatch-based screening for psoriasis-related cutaneous metabolites [35].

### 2.2. Atopic Dermatitis (AD) and Other Inflammatory Dermatoses

Ten papers were focused on AD or other types of dermatitis, the majority of them assessing cutaneous metabolites. Not surprisingly, histamine, urate, and serotonin were found to be increased in AD patients and are all inflammatory and/or itch mediators [43]. Metabolomic analysis of atopic skin also confirmed a dysregulation of skin lipid metabolism in AD with a relative shortage of several ceramide subclasses [37]. Moreover, several studies exploring metabolic changes and miRNAs involved in AD pathogenesis have also recently been published [22].

Multi-omics data confirmed increased glucose consumption and IL9-mediated redirection of metabolic flux towards lactate, with a subsequent reduction of tricarboxylic acid (TCA) cycle intermediates in HPKs [72]. Ilves and co-authors recently demonstrated increased levels of putrescine, dimethylarginine, acetyl-L-carnitine, glutamate, methionine, and sphingolipidsin atopic skin [49]. Such alterations in metabolite levels indicate inflammation, impaired barrier function, and susceptibility to oxidative stress in AD. Interestingly, the tryptophan metabolic pathway seems to also play a key role in AD, as specific serum variations in the levels of tryptophan-derived metabolites have been described in this setting [50,108].

Carnitines, free fatty acids, lactic acid, and other metabolites involved in energy metabolism have also been described to be increased in the AD population [48].

Not surprisingly, histidine—a precursor of histamine—was also found to be increased in an in-vitro model of allergic contact dermatitis [36].

Urinary metabolomes have been shown to be significantly modulated by specific agent exposure in the setting of AD: propyl-paraben has recently been demonstrated to be associated with aeroallergen sensitization and EASI score, mainly through the induction of metabolomic changes in specific metabolic pathways involving oxidative-stress response, mTOR, peroxisome proliferator-activated receptors, aryl hydrocarbon receptor signaling, and tricarboxylic acid cycle [13].

### 2.3. Melanoma

Thirteen publications were focused on melanoma and nearly all of them applied metabolomic techniques to skin samples. To date, metabolome analyses have mainly been aimed at identifying potential metabolic biomarkers for the diagnosis and the prognosis of cutaneous malignant melanoma [38,87].

Not surprisingly, the alpha-Melanocyte-Stimulating Hormone (α-MSH) metabolic pathway was found to be altered in melanoma cells [90]. Previous studies already hypothesized such a pathway to be an interesting therapeutic target and found melaleuca quinquenervia essential oil to efficiently inhibit α-MSH-induced melanin production and oxidative stress in melanoma cell lines [30].

A study from Abaffy and co-authors was aimed at comparing potential differences between normal skin, common nevi, and malignant melanoma through GC-MS and HS-SPME (head space-solid phase microextraction) [21]. Dodecane, 4-methyldecane, and undecane were identified as candidate markers of malignancy; these alkenes are involved in oxidative stress and membrane peroxidation [20]. The same group also identified lauric acid and palmitic acid as melanoma-specific volatile metabolites; increased lipid synthesis due to cell growth and proliferation in cancer and increased oxidative stress could explain such findings [19]. In 2020, Taylor and colleagues performed a metabolomics investigation of primary melanoma, metastatic lesions, and matched extra-tumoral microenvironment (EM) tissues [95]. Pathway-based results led to the identification of metabolic changes in ascorbate, aldarate, propanoate, tryptophan, histidine, and pyrimidine; such specific alterations were found to be present in both primary and metastatic melanoma but not in EM, therefore suggesting they are crucial in the initiation and/or maintenance of melanoma [95].

N-acylsphingosine-amidohydrolase-1 was demonstrated to promote melanoma growth and metastatization by suppressing peroxisome-induced ROS production in an in-vitro model of melanoma [71]. Other biomarkers indicating metastasis were identified by Kosmopoulou et al. through human melanoma-cell metabolic profiling; the authors indicated a critical role for purine, pyrimidine, and amino acid metabolism in the metastatic process [62]. Glycerophosphocholine levels were found to be reduced in exosomes derived from melanoma cancer stem cells and patients’ serum [83].

Untargeted NMR metabolomics of two mouse melanoma models labeled with 13C-methylmethionine were used to search for the NMR-visible set of cellular methyl acceptors denoting the global methylome [78]. Tumor models were B16 melanoma cell cultures and B16 melanoma tumors, which may be considered as two stages of B16 tumor development. Based on 2D 1H–13C NMR spectra and an orthogonal partial least squares discriminant analysis of the spectra, the study revealed markedly different global methylomes for the two melanoma models. The methylome of B16 melanoma cell cultures was dominated by histone methylations, whereas that of B16 melanoma tumors was dominated by cytoplasmic small-molecule methylations. A comparison of tumor models also exhibiting a differential expression of aerobic glycolysis provided clues to a methyl metabolism shift during tumor progression [78].

Finally, in 2019, Khosravi et al. identified 35 candidate drugs for melanoma treatment based on a combination of genome- and phenome-wide association studies, transcriptomics, and metabolomics [59].

### 2.4. Non-Melanoma Skin Cancers and Actinic Keratoses

Eleven papers focusing on non-melanoma skin cancer (NMSC), actinic keratosis (AK), and UV damage were retrieved from our search.

Most of the available data are based on cutaneous sample metabolomic assessment, with only one publication focusing on serum metabolites [41].

In-Vitro models of sun-damaged skin suggest that glycerophospholipid and glutathione metabolism are crucial for UV-damage response in the dermal compartment [101], while the pentose phosphate pathway and glycolysis are mostly impaired when both fibroblasts and keratinocytes are exposed to UV radiation and oxidative stress [63]. Even more importantly, in-vitro skin models of cancerization empowered the identification of metabolomic pathways that are crucial for tumorigenesis in general, with potential therapeutic applications in a broader oncological setting [93].

Several authors already reported on metabolomics as a useful tool for discriminating different types of cutaneous cancer [41,74]. Specific serum metabolites were found to differ between patients with squamous cell carcinoma (SCC) and melanoma, and both skin cancer groups could be distinguished from healthy controls [41]. SCC-specific metabolites included glycerol, 4-hydroxybenzoic acid, sebacic acid, fucose, and suberic acid. Moreover, glycolysis is notably a key metabolic pathway that is altered in both basal cell carcinoma (BCC) and SCC [47,74,79].

As for other diseases, metabolomics has also been used for the evaluation of therapeutic efficacy of new experimental molecules. Zhu et al., for example, recently described flavonoid derivatives to efficiently induce apoptosis and autophagy in in-vitro models of cutaneous squamous cell carcinoma [107].

Tumor progression from pre-neoplastic to neoplastic lesions has also been investigated from the metabolic point of view. Our group recently demonstrated a grade-independent model of squamous cancerization through the association of NMR metabolomic profiling and histopathological analyses [85]. We also confirmed an imbalance of the redox state and increase of skin metabolism both in SCC and in all-grade AKs as a metabolomic signature of sun-exposed skin [14,85].

Finally, the combination of metabolomics with other techniques, such as proteomics and transcriptomics, has also been used to confirm the presence of specific alterations in metabolic pathways and therefore validate metabolomic findings [105]. Such an integrative approach, for example, found an increase in glutathione levels, and a parallel decrease in fumaric acid in the setting of in-vitro arsenic-induced malignant cell transformation [105].

### 2.5. Wound Healing

Alterations of metabolic pathways in the wound healing process have mostly been investigated in murine models. The first publication focusing on cutaneous metabolomics assessed the use of expanders in minipig skin through NMR spectroscopy in the setting of plastic surgery. The metabolic profile of skin under stretch (inflated expanders) was found to be similar to that of control skin, while cutaneous samples from non-inflated expanders displayed increased anaerobic glycolysis and an altered energetic state, as demonstrated by higher creatine/phosphocreatine ratios [70].

More recently, D-series resolvins (derived from omega-3 fatty acids) and other related mediators were found to be involved in skin repair in animal models of wound healing, and their topical application has been proposed for therapeutic purposes [44].

Ashrafi and co-authors reported linolenic acid and glycerol to be significant metabolites, allowing differentiation between healthy skin and the earlier phases of wound healing—being particularly abundant between day 7 and 14. On the contrary, L-glutamine, a 1,3-dihydroxyacetone dimer, and adenosine were differentially expressed at later time points, suggesting their presence and abundance may represent a measure of varying wound maturity [10].

Several authors have also investigated the diabetes-related impairment of cutaneous healing. In particular, Sood and collaborators identified collagen synthesis, nitric oxide production, inflammation and fibroblast proliferation to be significantly impaired in diabetic skin [92]. The metabolic effects of a Chinese herbal remedy for non-healing wounds (Huiyang Shengji formula) have also been investigated in a rat model of a diabetic skin ulcer. Such substances could improve glucose and branched-chain amino acid metabolism and enhance antioxidant and pro-angiogenetic properties in diabetic skin [103].

A recent publication from MedStar Washington Hospital Burn Center identified burn-specific changes in murine sera and skin samples [24]. Skin changes affected inositol phosphate, ascorbate, alderate and caffeine metabolism, and the pentose phosphate pathway; such alterations were more delayed and less synchronous when compared to those detected in sera.

Moreover, external-agent exposure in the wound healing setting has been investigated through metabolomic techniques; graphene-related materials, for example, have been described to reduce the ability of HaCaT cells to heal wounds [40].

### 2.6. Other Dermatological Disorders

Several other dermatological conditions have recently been studied from the metabolomic point of view, including inflammatory infectious and neoplastic conditions [80,89,104,106]. Increased levels of L-glutamate have been found in skin samples of cutaneous T cell lymphomas and adjacent skin, together with decreased adenosine monophosphate [65]. Several alterations in fatty acid metabolism have been associated with Lyme’s disease [39,76]. De-novo fatty acid synthesis and the mitochondrial tricarboxylic acid cycle seem to have a deep impact on vitiligo pathogenesis [86]. Autoimmune connective tissue disorders have been investigated through metabolomic profiling, and very specific disease-specific markers have already been identified [27,104]. Moreover, rare genetic disorders (e.g., pseudoxanthoma elasticum) have been studied from a metabolic point of view, giving new insights into the pathogenetic pathways [52,84].

Finally, the metabolomic fingerprint of leprosy has also been widely studied [23]. Arachidonic acid, eicosapentaenoic acid, and docosahexaenoic acid have been demonstrated to be increased in the sera of leprous patients, while urinary metabolites were postulated to discriminate endemic controls from untreated patients with mycobacterial disease [73].

### 2.7. Skin Function

Various studies aimed at better understanding skin function and composition have been published so far. In this setting a metabolomic approach can prove the reliability of in-vitro models of skin or of other tissues and/or biological fluids [28]. Untargeted high-resolution metabolomics, for example, has been used to assess the metabolome of suction blister fluid as a surrogate of interstitial fluid [81].

Metabolite changes have been detected so far in various physiological processes involving the skin, including melanogenesis, aging, and even death [32,64,77].

Metabolomic techniques have also been employed in the evaluation of skin function and external agent-induced skin changes. In this setting, drug-induced cutaneous toxicity has been largely studied.

Jiang et al. investigated the effects of polycyclic aromatic hydrocarbons exposure on skin in vitro [53]. A complete metabolome assessment was performed after HaCaT-cell exposure to phenanthrene. Interestingly, amino acid, glutathione, and glycerophospholipid metabolism emerged as being impaired, resulting in a reduced antioxidant status.

Furthermore, the effects of low-dose radiation were studied using an in-vitro 3-D human full-thickness skin model, which led to the identification of specific molecular pathways involving oxidative stress, nitric oxide signaling, and transcriptional regulation through the SP1 factor [96].

More recently, a Chinese group focusing on metabolic changes in arsenic-induced skin lesions found a negative association with tryptophan and phenylalanine levels.

Furthermore, such amino acids alone were reported to be a potential tool for distinguishing patients developing skin lesions after arsenic exposure from healthy controls, therefore potentially being used as early disease markers [98].

A relatively common application of metabolomic techniques is the scenario of topical products. For example, NMR spectroscopy has been efficiently employed for assessing silver-nanoparticle-induced skin toxicity [29]. The Portuguese group found both silver nanoparticles and H_2_O_2_ to induce the downregulation of glycolysis and energy production; however, some metabolic pathways (GSH synthesis, glutaminolysis, and the Krebs cycle) were described to be specifically altered by the application of silver nanoparticles, independent of ROS-mediated mechanisms. Such results pave the way to metabolome assessment for the screening of pre-clinical toxicity of nanomaterials.

With regards to this, a recent paper from Jacques et al. describes the use of metabolomics to assess the biological characteristics of a candidate cosmetic ingredient compared to a structurally similar reference compound [51].

Triclosan-induced dermal toxicity has also gained interest lately due to the widespread use of such antimicrobial agents [68]. In fact, Triclosan exposure has been reported to induce purine and glutathione metabolism, with parallel downregulation of amino acid metabolism and keratinocyte lipid-metabolism impairment. A metabolic biomarker analysis revealed ROS and ammonia overproduction to trigger inflammation and cell apoptosis in the skin, as was also confirmed in HaCaT cells [68].

Treatment with a stearoyl-CoA desaturase 1 inhibitor was described to induce specific metabolic changes in sebaceous secretions [58]. With this aim, NMR was used to monitor drug-induced sebaceous gland atrophy in rodents and was therefore proven to allow non-invasive assessment of lipids in sebaceous excretions. Other studies focusing on sweat metabolites led not only to deeper knowledge of axillary skin metabolism and odor generation, but also to the identification of site-specific metabolites in different cutaneous locations [42,46].

Moreover, non-invasive approaches for cutaneous metabolome assessment have also brought new perspectives in the setting of anthropophilic mosquito–host interactions, and to the potential identification of attractants or repellants for vector control strategies [100].

A very recent publication on the effects of pollution on human skin clarified the molecular bases of clinical worsening of various inflammatory and non-inflammatory dermatological conditions (including eczema, acne, lentigines, and wrinkles) after environmental pollutants exposure.

Metabolomic assessment led to the identification of 350 metabolites, 143 microbes, and 39 polycyclic aromatic hydrocarbons that correlated with pollution exposure [75].

Due to increasing interest in antioxidants in UV-damage response, the effects of short-term (7-day) supplementation of green tea extract (GTE) on skin metabolomes have been investigated in mice models. GTE supplementation helped the skin metabolome defend against UV stress [54]. From combined skin and gut metabolome assessment, green-tea-induced changes in gut microbiota (involving Bifidobacteria and Lactobacillus spp in particular) were found to be responsible for positive effects on the UV stress response, therefore confirming the importance of the gut–skin axis [54].

## 3. Materials and Methods

A search was conducted in the MEDLINE, EMBASE, Cochrane, and Scopus electronic databases from inception to present. The detailed search strategy for MEDLINE (PubMed) used the following terms: dermatology [Title/Abstract]) OR skin [Title/Abstract] OR cutaneous [Title/Abstract] AND metabolomics [Title/Abstract] OR metabolite [Title/Abstract]. The terms were adapted for the other databases as appropriate. All the major journals were indexed. Only journal articles were taken into consideration, while books and book chapters were excluded. Articles without full text electronically available and/or English translation were also excluded. Only journal articles focused on metabolomics and skin were considered, excluding those that focused on skin microbiome (bacteria, fungi, viruses). Review articles were not included. Only studies on dermatological disorders, skin physiological functions (e.g., changes of skin metabolome in response to external stimuli), or regarding new methods for skin metabolome assessment were included in the present study. The search was not restricted on specific metabolomics techniques (e.g., NMR, GC-MS, LC-MS), type of sample (cells, biological fluids, cutaneous tissues), or biological source (human, animal).

The following data was collected for each paper: author, journal, year and type of publication, method used for metabolite assessment, target skin disease, type of biologic sample, study type (ex vivo/in vivo), subject enrolled (human/animal), number of cases and controls, type of analysis (qualitative/quantitative), exposure to external agents and/or pharmacological treatments, relevant metabolites, and metabolic pathways involved. A Student’s t-test was used for comparison of two different variables. P-values lower or equal to 0.05 were considered statistically significant. No further statistical analysis was performed due to the heterogeneity of the data and methods used for metabolite assessment.

## 4. Conclusions

Metabolic profiling has been largely applied for the characterization of physiological skin functions and for the discovery of metabolic changes driving disease onset and progression in the dermatological setting [109]. Advancement of analytical techniques has already shown promising results in enabling complete metabolome quantification using very small volumes of sample. New non-invasive approaches for metabolome assessment will surely provide easier and faster tools for the identification of disease related and will pave the way for the routine use of such assays in the clinical setting.

## Figures and Tables

**Figure 1 ijms-23-08776-f001:**
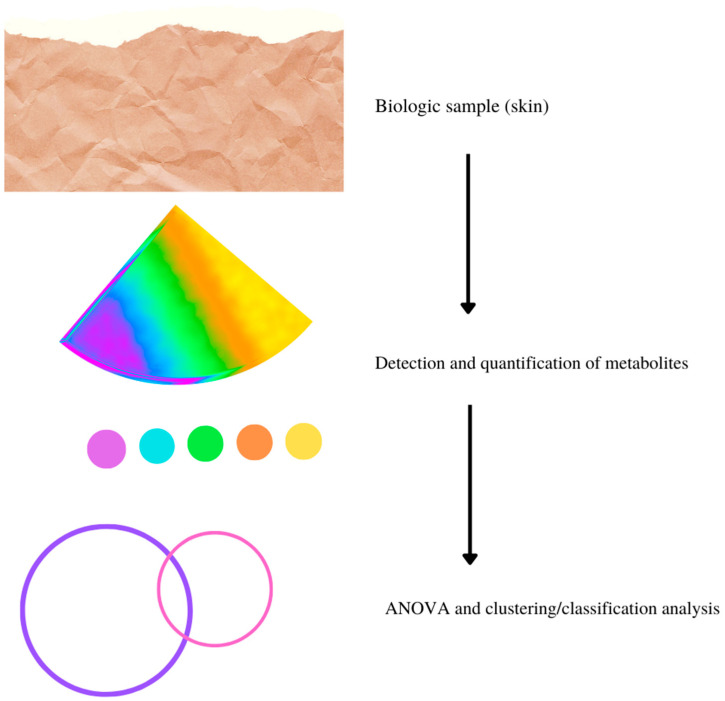
Typical workflow of metabolomic assessment in dermatology.

**Figure 2 ijms-23-08776-f002:**
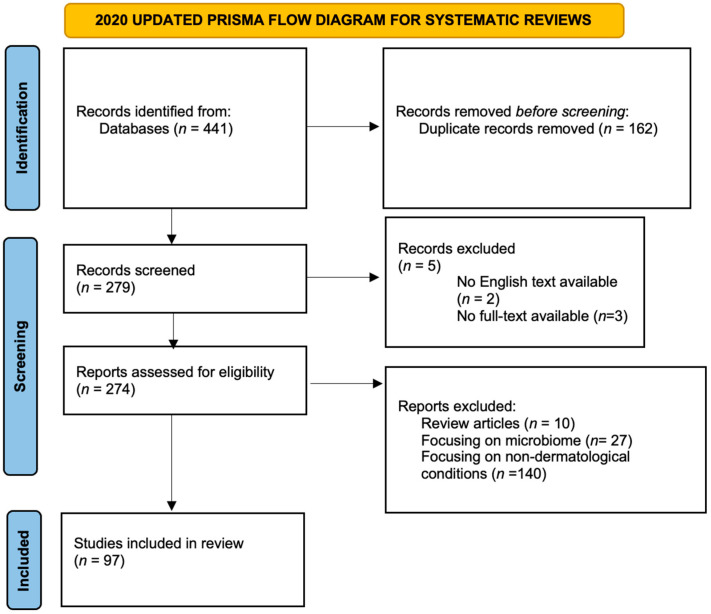
PRISMA flow diagram according to 2020 updated guidelines for systematic reviews.

## Data Availability

Not applicable.

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
