# Peer review of "Current Knowledge in Skin Metabolomics: Updates from Literature Review"

_ijms, 2022, doi:10.3390/ijms23158776_

Round 1
Reviewer 1 Report
Dear Authors,
The manuscript entitled "Current Knowledge in Skin Metabolomics: Updates from Literature Review " offers a molecular point of view of the different cutaneous diseases such as psoriasis, atopic dermatitis, melanoma, and non-melanoma skin cancers, among other. The metabolites are also a useful tool to physiologically characterize the skin.
This is a very nice manuscript, which is well-written and well-organized and the summarized information about the metabolites and biomarkers provided is very valuable to understanding the molecular bases of the cutaneous diseases considered in the article and may lead to motivate further investigations in the field of diagnosis and even target new specific treatments.
I only have two comments:
- line 250 SCC-specific metabolites [...], the authors provide the description of SCC in line 253, it will be helpful for the reader to have the description in line 250, at its first appearance.
- line 344: H2O2, please add the 2 as the subscript format.
Kind regards,
Author Response
Dear reviewer,
thank you very much for your kind comments.
According to your suggestions we made the following changes:
- the acronym SCC has been explained at its first appearance
- 2 has been added as the subscript format in H2O2
Reviewer 2 Report
Metabolomics in the future may be of significant help in the diagnosis and evaluation of clinical activity of many diseases. Currently, there is a great interest of researchers in metabolomics in many diseases. The authors presented an interesting review on the importance of metabolimics in skin diseases. The methodology for the selection of literature in the presented review is correct. The literature used is up-to-date. The text is basically well written.
Minor comments:
1. line 158-160 - the authors should extend the information on metabolomics in comorbidities in psoriasis - the very general statement given is insufficient.
2.line 280-285 - the text should be supported with citation - I guess it's position 10 in the literature
3. similarly, for the reader's convenience, authors should include citation numbers for the studies included in the table.
Author Response
Dear reviewer,
thank you for your kind suggestions.
We made some changes in order to improve the quality of the paper:
- the part on metabolic changes in psoriasis comorbidities has been expanded
- lines 280-285 have been referenced
- citation numbers have been included for the studies in the table